# Husserl and Heidegger on Modernity and the Perils of Sign Use

**Johan Blomberg**

Centre for Languages and Literature, Lund University, Box 117, 221 00 Lund, Sweden; johan.blomberg@semiotik.lu.se

**Abstract:** In his late writings Husserl emphasizes how the semiotic properties of writing, and of mathematical formulae and diagrams, are crucial for the historical, cross-generational survivability of meaning and specifically indispensable to the operation of scientific knowledge. However, the demand for objectivity, exactitude, and repeatability insidiously interferes with the meaning that such signs seek to express. This leads to a duality of objectivity encapsulated in the notion "the sedimentation of meaning". On this view, the transmission of objectivity established in an original sense-constituting act cannot survive unless being deposited in some external form, which at the same time risks the original sense being irrevocably lost in a web of signification that amounts to nothing more than empty and meaningless symbol manipulation. I discuss Husserl's analysis and propose that it is limited by its one-sided focus on the negative impact of modernity. I compare Husserl's account with Heidegger's even more radical critique of modern society as one where a so-called "technological" mode of "revealing" reigns supreme at the expense of eradicating other, and more authentic ways to apprehend the world. I critically reconstruct the respective position of both thinkers and show how they point not only to a criticism of the instrumentalization and formalization of knowledge in modern society, but that they are just as importantly highlighting essential semiotic properties of signs.

**Keywords:** sedimentation; philosophy of technology; writing; representations

## 1. Introduction

In their respective analysis of modernity, both Husserl and Heidegger offer a bleak diagnosis of their present times [1–3]. Their view is to a large part motivated by what they perceived as an all-encompassing instrumental scientific-technological rationality that with its strict demands for exactitude, objectivity and control is at risk of eradicating the possibility for a philosophically authentic, ethically responsible, and meaningful existence. What arguably sets apart Husserl and Heidegger's respective reading from other early 20th-century critiques of modernity is the philosophical analyses that underlie the vision of an existential threat embedded in the very structure of modernity: it harbors the ultimate risk of losing the ground that allows asking questions concerning the kind of experiential meaning that serves as the ultimate foundation for any type of knowledge. Their critique of modernity and their ultimate outlook are considerably different and this should not be neglected. I nevertheless emphasize that they share some strikingly similarities in their assessment of the modern world as one that risks warding off access for critically engaging with questions concerning origins and ultimate meaning, and the importance of a cautious, reflective and responsible philosophical scrutiny of the preconditions that allowed such a pre-apocalyptic state to emerge. Where they differ is in the root causes and, as we shall see, in their response to how the ailment can be overcome.

Even though both Husserl's and Heidegger's analyses of modernity have been largely influential, with their similarities and differences widely discussed [4–6], what has been somewhat neglected is how essential the role of language and other semiotic resources (such as diagrams, mathematical formulae and abstract models) are for the predicament of modernity and the need for a responsible attitude towards the way semiotic resources

might risk intervening with the sense they seek to communicate. In this paper, I aim to show how their respective analyses–without in any way trying to make the case for a decisive similarity where their accounts coalesce into a harmonious whole–are intrinsically connected with specific conceptions of what can be represented, especially in a formal manner, and how an identical sense can be transmitted throughout time and space. One could perhaps even say that modernity turns out to be–by and large–a semiotic crisis, but not primarily in the sense noted by for instance later poststructuralist thinkers such as Derrida [7][1]; rather, the crisis comes from a particular semiotic attitude embedded particularly in modernity that involves forgetfulness and carelessness for the preconditions that truly and authentically endow signs with sense. At the same time as Husserl and Heidegger in radically different ways fundamentally question the (mis)use of semiotic resources, they simultaneously both point to the intrinsic function of the same resources in the formation, establishment and transmission of meaning. To show these two sides of their critique, I focus specifically on the central Husserlian concept of *sedimentation* [2] and Heidegger's account of *technology* [3], which both show deep affinities with semiotic matters. By reading them through such a lens, I also propose that the problem they locate is not necessarily limited to a particular historical epoch. Instead, they are also (and perhaps contrary to their intentions) discovering general formal features of signs. What modernity arguably brings to the table is the ability for a plethora of novel semiotic technologies with an exponential acceleration of their dissemination.

I begin in Section 2 with summarizing Husserl's notion of modernity as a crisis based on a loss of meaning most clearly exemplified in the fragmentation and instrumentaliza­tion of the sciences. Section 3 shows how this crisis is ultimately connected to historical semiotic processes of formalization and 'technization' [1] where sense is made objective but ultimately hollowed. Sections 4 and 5 turn to Heidegger with the similar structure of explicating his assessment of modernity before turning to the way a modern understanding of the nature as an exploitable resource is connected to a belief in fully capturing objects in exact and precise "representations" (*Vorstellungen*). I end in Section 6 by comparing and critically examining their respective assessment of modernity, focusing specifically on the constitutive function attributed to representational technologies in the formation of modernity.

## 2. Signs of the Crisis

### 2.1. Modernity and the Crisis

The modern age is one of unparalleled scientific and technological advancements that fundamentally has transformed–and continues to transform–the world we find ourselves in. It is common to locate the basis for these revolutionary developments in a mindset specific to naturalist inquiries, which allows for objectively true knowledge rather than for pure speculation based on authority and tradition. As we shall see, an operative word in both Husserl's and Heidegger's criticism of such an interpretation is the visual metaphor of "concealment", e.g. [1,3,8][2]. While they come from very different places and place their emphasis on different aspects of modernity, both argue that particularly natural science and modern technology hides its own epistemological presuppositions which makes these achievements seem unquestionable and taken for granted. This embedded concealment at the roots of a naturalist understanding does however not only lead to a progressive improvement but at the same time to a loss of the meaning that motivate and provide such an understanding with validity in the first place. A key part in this simultaneous progress and deprivation is a growing reliance on various types of signs, most clearly exemplified by mathematical symbols and diagrams, which, albeit objectively accurate, are formalized and abstracted to the degree that they hollow out and gradually displace the original meaning that endowed them with sense in the first place. Exactly what such an authentic form of meaning amounts to is, as well shall see, a point where they fundamentally differ.

I use the term "modernity" to point to a historical epoch singled out by both thinkers. While we find terms like "modern idea" in Husserl's work (e.g., [1], pp. 12, 14, 21, 65)

and "technological age" characterized by "machine technology" as "the essence of modern metaphysics" in the latter (e.g., [3], pp. 21, 34, 116), I find "modernity" to be an appropriate term for what they have in mind. However, one should exercise caution to *not* just identify their respective understanding with industrialization, technological-instrumental rationalization, increasing bureaucratic control, urbanization, the birth of (modern) science or other features usually associated with modernity. More relevant in the present context are novel philosophical approaches (some highlighted by Husserl are naturalism, empiricist skepticism and psychologism [1,9,10] whereas Heidegger rather tends to emphasize Cartesian rationalism, transcendental idealism, humanism and nihilism [3,7]). In this regard, they both argue that the rise of modern science is tightly interwoven with both technological advances and philosophical enterprises that fundamentally transform our encounter with the world and one another. This is for Husserl specifically based on a conceptual shift where a naturalist and calculative gaze of scientific reason replaces other forms of understanding, especially one that seeks to explicate the phenomenological foundation that endows the quest for truth and knowledge with its meaning in the first place [10] (p. 8). This would have the further existential significance of depriving us of what "make[s] possible a truly satisfying life" [10] (p. 5). As the compartmentalized and fragmented sciences and the technologies it gives rise to are implemented to an increasing degree, the validity for even asking questions about the phenomenological origin for science in human experience is also annulled[3], which Videla summarizes as follows: "once such an objective understanding had done away with the contents of the experienced world, the issue of meaning–the questions of reason–could no longer be raised" [11] (p. 191). With this brief discussion on modernity in mind, let me know turn to Husserl's concept of crisis.

### 2.2. Crisis: The Forgetfulness of Meaning

In a number of lectures held in Prague and Vienna during 1934 and 1935, Husserl presented his diagnosis of contemporary European science and philosophy as being in a state of "crisis", one with deeper roots in the culture of his time [1]. These investigations are carried out in a way that echoes his phenomenological dictum "back to the things themselves", not only in its reading of intellectual history but also in its critical self-examination of Husserl's earlier philosophy where the role of history was downplayed and explicitly rejected as a theme for phenomenological inquiry [12] (p. 73f). In this paper, I will not primarily discuss the importance of history and temporality and their specific articulation within Husserl's late thought (though, see for instance [13,14]), but rather limit myself to Husserl's view on modernity as offering a radical transformation brought about by the scientific revolution, and how this entails a fundamental conceptual shift in which world is considered primary: the world of natural science in favor of the life world (*Lebenswelt*). I emphasize this part of the crisis since Husserl argues that this momentous rupture is in part upheld by technologies like writing and the type of status that is attributed to them[4].

In a sense, the critique of modernity found in Husserl's late works is reminiscent of his earlier criticism of ignoring the indispensable participation of transcendental subjectivity for knowledge (e.g., [9]). However, at the time of writing *Crisis of the European Sciences*, the emphasis has shifted from a critique of an empirically oriented psychologism into a fear of losing the original experiential life world, which is what irrevocably gives birth to *any* kind of meaning–including science. Using the dictum "to be is to be measurable" often attributed to Galileo as the radical expression of a naturalist attitude, Husserl argues that modern science assumes the world to be mathematically measurable. Instead of departing from the life world, such a naturalized scientific worldview would assume that what is objectively quantifiable has ontological priority. To see such a perspective as basic and as presenting us with ultimate truth thus presupposes a radically transformed, but according to Husserl, fundamentally mistaken epistemological starting point where "we take for *true being* what is actually a *method*" [1] (p. 51).

In his expose, Husserl takes Galileo as a metonymic symbol for modern science who "at once [is] a discovering [*entdeckender*] and concealing [*verdeckender*] genius" [1] (p. 52). The mathematization of nature that has taken place in modern science serves as a double concealing cloak, which is a process that could be exemplified by looking at the historical transformation of space. First, a modern scientific conception of space (with properties like three-dimensional, completely homogeneous, infinitely extendable in all directions, etc.) hides not only its historical but its phenomenological origin in an embodied or lived space comprised of concrete bodies of various physical shapes to which we are related through the concrete activities of the living body and its position in a space relative to it [15,16]. These spatial relations are experientially determined relative to how they are given to the living body as an "ever-abiding point of reference" (*immer bleibende Beziehungspunkt*) ([15] (p. 66). This living body, in turn, can hardly be separated from the life world which serves as a horizon for all of its activities. Husserl proposes that an incremental process of abstraction from these concrete bodies led to a practically oriented proto-geometry aimed at manipulation and control of the concrete environment, as in for instance land surveying [1] (pp. 27–28, 49)[5]. At the stage of proto-geometry, Husserl speculates that generalized shapes still "bound" to concrete bodies like 'round' were used. As the process of idealization continued, the link to material entities becomes weakened and ultimately severed. This leads to "pure geometry" as a rational science concerned with ideal shapes governed by formal mathematical relations (like the mathematical constant $\pi$ or the Pythagorean theorem)[6]. While they originate from physical shapes, geometrical forms are so much abstracted that their mode of givenness must be considered as distinct from their generative-phenomenological origin in lived space (see [17] p. 267f for Husserl's discussion on how these two type of idealities differ in terms of either being "bound" to material entities or "free" from such instantiation). This is manifest in idealized space exhibiting features that cannot be found in lived space, such as complete uniformity, exact measurability, external to a human perspective and infinitely extending in all directions (as in for instance Descartes' characterization of space in *Discourse on the method*). Ideal space thus hides its origin as an "achievement" originating from life world.

The momentous rupture of modern science is the second step of concealment: to apply such a conception of space back onto the life world. As this occurs, this world is no longer considered to be basic; rather an objectified conception of physical space emerges as something that can be precisely measured and quantified. This conception sees space as a "huge block" [18]. Through such a process, the life world and the necessary contribution of subjectivity even for science become completely obscured. This is not to say that Husserl wants to claim that geometry as the science of "pure space" [1] can be reduced to its preconditions in the life world. As a rational and axiomatic science, geometry is characterized by operating with ideal entities such as circle, lines, etc. with formally defined features. Once instituted, this means that their ideal sense as unchangeable and everywhere applicable as exactly the same attains independence from its origin in the perceivable shapes encountered in the life world [2] (p. 357). In doing so, Husserl attempts to maintain both the integrity of ideality and simultaneously insisting on the phenomenological necessity of its origin in the life world. This is summarized by Hopkins as "the inseparable connection between the meaning [*Sinn*] proper to the ideal a priori that is the defining characteristic of objective knowledge and the historicity of this meaning's origination" [19] (p. 180). This connection is decidedly one of the most complicated issues in Husserlian phenomenology, which we will have reasons to return to in Section 3.

The double act of idealization from lived space and technized objectification back onto lived space captures some of the essential features of the crisis: the compartmentalization of science from the life world, clearly expressed by the strict division between objectivity and (inter)subjectivity that in turn invites skepticism and ultimately the loss of an experientially grounded meaning. This occurs as a mindlessly calculative rationality starts to live its own life as a conquering automaton that submits increasing swaths of the life world. I return to this in more detail in Section 6, but one pertinent expression of this attitude in

our time is the type of quantification procedures common in New Public Management. As noted by for instance Espeland & Stevens [20], the deployment of such strategies does not just serve as measuring or overlooking the activity of social institutions, but to actively participate in shaping what such institutions are. Peter Woelert summarizes how this dual cloaking of the life world is operative in modernity: "the rationalisation of space, despite having its own historical genesis, gains a certain automatic momentum, a certain foreclosing pre-determining force, once a particular idealising rupture or crisis has taken place" [21].

The increasing distance between direct experience and formalization does not just entail a rift between the life world and the modern conception of science but actually amounts to a crisis of civilization, given Husserl's view on the place of science and philosophy as a crowning achievement of "European" culture. The present state can ultimately be traced to how science and modern culture has lost track of its "inborn teleology" (*eingeborene Teleologie*) [1] (p. 273), which Husserl finds as the catalyst inaugurating Greek philosophy. This faith in a responsible and universal reason has been replaced by overreliance on a naturalistic approach that in turn has given rise to a backlash in terms of a "skeptical deluge" (*skeptischen Sintflut*) [1] (p. 14) (cf. Section 6 below). Husserl claims that the birth of Greek thinking marks the initiation of a unique culture with a specific "spiritual" (*geistig*) telos located in the continuous and arduous struggle for truth based in universal reason [1,10]. The historical particularity of a certain life world is thus transgressed by the search for truth not limited to a life world-relative horizon (see [12] for a discussion on universal truth vs. historical particularity in the context of Husserl's concept of life world). Through history, such cultural traits have been handed down as a tradition that Husserl calls "European science". Situated within a life world, every generation necessarily becomes "heirs to the past" [1] (p. 17)[7], which forms the basis for the *generativity* of the life world: the fundamental world of meaning as both upheld through the ongoing activities and handed down as tradition to the coming generation. The search for an ultimate, incontestable truth is one such feature that Husserl identifies with the European life world. Even as we venture forth on this arduous task, a scientist or a philosopher never starts over from the beginning but continues and builds from previous findings in an ever-expanding process driven by this telos for seeking truth[8]. This is arguably also the case when aiming to start over from a new and incontestable ground, insofar as such a revolutionary aim occurs precisely against the background of a tradition [12] (pp. 81, 82). This same process, albeit necessary for instituting a progressively ongoing scientific tradition, is at the same time at risk of eradicating the stable ground of subjectivity required for the constitution of objective knowledge [2] (p. 362f).

The momentous shift where the mathematically calculable world is taken to be ontologically primary entails a transformation of the ultimate philosophical quest for truth, which in turn means erasing the very foundation on which it is built. As this occurs, mathematical understanding of nature transcends its limits as a method and turns into a metaphysical reading, as stated by Carr:

> Mathematical science is a method which considers the world *as if it were* exclusively a manifold of measurable shapes; the ontological interpretation simply states that it *is* such manifold. Now the scientific problem is different from the philosophical problem: the first seeks intersubjectively exact knowledge *about* the world, the second hopes to determine the true nature of reality. But here the solution to the first problem is taken as the solution to the second and a hidden shift of meaning has occurred. All subsequent problems connected with the world–its scope, its beginning and end, man's place in it and, above all, his knowledge of it–henceforth operates with this conception of reality as a presupposition. [12] (p. 86)

Knowledge becomes held in a suspended state of strict adherence to a rigid procedure for producing more and more empty calculation. Surely there will still be discoveries and advances internal to such a "technization" of nature [1] (p. 46f), but with the bond

to the life world completely severed these will hardly amount to anything more than automatic manipulation of symbols without them having any meaning outside of their instrumentalization. Since Husserl adamantly wants to retain the validity of the sciences and provide them with a transcendentally stable ground, he seeks the solution to the crisis in a phenomenological method of unearthing the sedimented layers that made science (and arguably modern culture more generally) possible on the basis of transcendental subjectivity and its place in the life world, i.e., where disciplined calculation and rigid adherence begin to take form in the first place. The task set forth in Husserl's late writing is thus one of "restoring" and "shaping sense anew" [10] (p. 10), which he describes with terms like "return-inquiry" (*Rückfrage*) [1] and "radical sense-investigation" (*radical Besinnung*) [10]).

Husserl's criticism of modern science might seem like an expression of a cultural conservatism resistant to the progression of knowledge and its positively transformative effects on society at large. Instead of applauding the breakthroughs that could be argued to improve human life, the cultural conservative might be seen as overexaggerating the risks of a soulless fact-informed and technocratically governed society. Such a reading, however, does not fit with Husserl's intentions. What is at stake is how the structures and processes that enable the functioning, dissemination and continuity of scientific prosperity is at the same time detrimental for the pursuit of truth and thereby catastrophic for a responsible and meaningful rationality. It is to this interplay we now turn.

### 3. Signs in the Crisis

Husserl returns time and time again to how and what type of meaning signs can express [2,9]. With his concern in finding the foundation for knowledge, the functioning of signs like mathematical symbols, diagrams and words takes on a specifically important role. These are, as Husserl argues (e.g., [1,2]) required both for the proper functioning of science and for performing mathematical calculations. It would hardly be possible to have mathematics without relying on some form of symbolically mediated thinking. At the same time, such signs have a formal character which quite clearly make them "emptied of meaning" (*Sinnentleerung*) [1] (p. 46). Husserl therefore aims to elucidate exactly how these symbols can have phenomenological validity in the first place. This is aptly described by Hopkins [19] as an attempt to "de-sediment" the many layers of meaning that have been built on an assumed transcendental foundation in intentional experience[9].

The problem of how signs participate in the formation of knowledge takes on a specific articulation in Husserl's late philosophy. In his critique of modernity, he argues that signs can be instrumentally operational while retaining only their procedurally formal function. Specifically, his earlier analyses are complemented by developing two interrelated points that grant a stronger emphasis on the capacities of language[10]. First, language is described as a cultural repository for sharing meaning and for letting in principle anything come to be known through its intersubjectively shared linguistic form. An indispensable condition for ensuring the survival of an intellectual tradition resides in the dissemination of knowledge where language and "the human world" are–as Husserl writes–"inseparably intertwined" (*verflochten*) [2] (p. 359). Second, at the same time as language is a requirement for knowledge to be shared, Husserl also sees a risk in how the "seduction of language" [2] (p. 362) allows for a "passive taking-over of ontic validity" [2] (p. 364). The establishment of stable meanings in language make it possible to operate at a surface level of the emptied signs without having to activate the whole chain that provided it with sense. The effect that has occurred with the reversal of which world is primary is that the "surrogative" signs replace the meaning they supposedly stood in for. This contributes to the crisis as one that "permits the total detachment from the originary acts and thus allows for a science without experiential ground" [22] (p. 85).

These two themes are central to the influential text *Origin of geometry* [2] (a posthumously published text where the original manuscript from 1936 is usually appended to later editions of *Crisis*). Husserl argues that objectivity–even in its most ideal form as found

in for instance rational disciplines like geometry–crucially perdures over time by being deposited in written form. As suggested by Blomberg [22], Husserl posits that written language allows for the preservation of sense across acts of transmission by providing two complementary dimensions: (I) ideal sense survives in language because it is comprised of ideal forms (i.e., words are ideal entities insofar as their meaning remains the same across the individual and contextually variable instantiated uses, see [2] (p. 357), [10] (p. 20) (II) by letting the ideal meaning be materially preserved in an external form. Only then can truth survive *independently* from the individual subjects, whose eventual demise threatens the availability of the knowledge they harbor. It is important to not misread Husserl on this point: he does not claim that the ideality of geometrical figures and formulae is the same as their graphical or mathematical representations [2] (p. 357). What is at stake is rather how to guarantee their ideality as exactly the same on all subsequent repetitions. To ensure invariance across each factual repetition, Husserl argues that even ideal object must be deposited into the life world in an externalized form that can be handed down and spread throughout history (see also [19], p. 83)[11].

The strong insistence on such a connection between writing and ideality clashes with some of Husserl's thoughts on science and truth. Within the community of scientists, the establishing of an objective truth entails its general acceptance and further availability for all practitioners. This means that there is no longer any need to "run through the whole immense chain of groundings back to the original premises and actually reactivate the whole thing" [2] (p. 363). The mathematician of today does not have to reignite its entire history in order to operate with mathematical truths and formulae. The previously made acquisitions form a depository from which any contemporary mathematician can draw. Over the course of time, the very self-evidence of a geometrical truth thereby becomes unnecessary for the practical functioning of mathematics. However, these symbols that uphold objective validity for a scientific discipline are themselves not the truths they stand for. Thus, in modern science there is possibility to preserve the experientially authentic originary disclosure that underlies the establishment of these sciences. They sever their connection to the life world, which means that the world conveyed by science is assumed to be the ultimate reality [1] (p. 23f). However, what occurs is that the forgetfulness of the origin makes scientists operate with signs and formulae "emptied of meaning". As it happens, this is also a root cause of the crisis: scientists manipulate formalized symbols to come up with more and more refined calculations, but despite their formal nature they have gradually been replacing the life world from which they originated that ultimately provides them with meaning and validity. In this way, Husserl's concern is not a conservative nostalgic lamenting of a modern world moving too far away from its own tradition, but at stake is the possibility of depriving ourselves of original meaning.

At the same time as Husserl sees the systematization and externalization of truth in written form as detrimental, he also finds this to be the litmus test for objective validity and a presupposition for a functioning science. As discussed above, Husserl argues that language participates in maintaining ideality but he does not account for *how* language may have such a preserving and even contributing function. Blomberg [22] suggests that *Origin of Geometry* introduces four distinct features of language without clearly distinguishing their role from one another: (a) language as collectively and intersubjectively shared; (b) the fixation of sense as a communal effort, (c) writing as allowing for preservation and communication across space and time, (d) rigorously defined scientific terminology which provides these terms with an objective sense that remains available as exactly the same on all uses, but this process of increased formalized symbolically mediated objectivity is also what hollows out the meaning that endowed them with sense in the first place.

Husserl thus displays an ambiguity towards the role of written signs and their functioning for the objectivity required by any valid scientific enterprise. On the one hand, they are indispensable for the functioning of science *qua* science and hence for the transmission and formation of objective knowledge. To achieve the essential trait of science as an endless *telos*, one cannot constantly unearth all the hardened layers of sedimentation

that comprise the history beneath the present state of knowledge. On the other hand, it is writing that lets us operate "at the surface", thereby irrevocably losing the connection to the ultimate foundation of knowledge in originary experience. It is for this reason that losing the connection back to the inception amounts not just to ignorance in face of the historical circumstances for what Popper called the "context-of-discovery" [23]. At stake is the risk of irrevocably losing what made an authentically objective sense possible in the first place (in a phenomenological reading of objectivity). The institution and transmission of the objective sense needed for science would thus seem to involve an interplay between forgetfulness and objectivity, which Derrida summarizes as: "creation which discloses and sedimentation which covers over imply each other" [24] (p. 118). When discussing Heidegger's notion of truth in the next section, I return to this relation between discovery and concealment.

The relation between written tradition as simultaneously both liberation and bondage in Husserl's analysis seems to invite the interpretation that it is not just a *particular* problem of modern Western culture, but an issue that befalls sense transmission *in general*. Such an interpretation is endorsed both by Buckley [4] and Derrida [24], which in its most radical formulation would take sedimentation of sense to be a precondition that applies *even on the first mention*. For sense to take linguistic form presupposes the effects of sedimentation as a structural possibility. According to the principles established by Husserl, the destabilization of sense and forgetfulness of the origin it entails belongs to the structure that renders the communication of a stable and well-defined sense possible. At first glance, to view this instability of linguistic–and particularly written–sense transmission as an indispensable precondition for the survivability of sense might seem like an anti-Husserlian position. I do however find that there is room for an interpretation compatible with a generative phenomenology of language. I say so for two different reasons, first: in *Origin*, Husserl regards sense preservation as necessarily a communal and intersubjective responsibility. In this way, the tension of written impersonal documentation as both liberating and threating sense should not necessarily lead to the nihilistic conclusion that it is impossible for sense to be communicated in an unambiguous and clear way; on the contrary, it is a structural condition to take responsibility for. Second, Husserl consistently points to the constitution of sense as necessarily incomplete. Even in the case of a perceptual givenness, one can always take a different perspective on the same material object. The adumbration of profiles that build up the complete object make the constitution even of perceptually given objects inexhaustible. To account for this, Husserl evokes the notion of "horizon" to indicate the–at every moment–conceived limit to our explorations. But if we were to ever reach this endpoint, it has—just as the horizon—moved further and further away. In a similar manner, one could build on Husserl's discussion to consider linguistic meaning as also constituted by a similar type of horizon.

## 4. Modernity and the "Essence" of Technology

### 4.1. The Forgetfulness of Being

Heidegger's entire philosophical project has been described as "crisis-philosophy" by Buckley [4] and as "apocalyptic" by Carr [11]. In this regard, his view radically differs from Husserl's belief in a telos common to all sciences with philosophy as the highest among them. According to Heidegger, there is a common root at the origin of philosophy and science that sets them, and arguably all subsequent thinking, off on a specific path to arrogantly and foolishly seeking domination over the world. In his later philosophy, Heidegger argues that this has reached its ultimate apex in modernity, which through its incomparably influential force restricts our understanding of what exists to that which can be calculated, predicted and thereby controlled by us in advance. The following quote from *The end of philosophy and the task of thinking* clearly articulates how different Heidegger's reading of philosophy is from Husserl's.

> We forget that already in the age of Greek philosophy a decisive characteristic of philosophy appears: the development of sciences within the field which phi-

losophy opened up. The development of the sciences is at the same time their separation from philosophy and the establishment of their independence. This process belongs to the completion of philosophy. Its development is in full swing in all regions of beings. This development looks like the mere dissolution of philosophy, and is in truth its completion. [25] (p. 57)

There has always been a potential embedded in the fabric of (Western) thinking that has reached its full mature actualization in modernity. While Husserl emphasizes sedimentation as an effect of losing touch with an authentic and responsible rationality, Heidegger rather makes the case that the preconditions for having such an understanding should be sought in a particular mode of human understanding, or "revealing" (*Entbergen*). The distinctive feature of modernity is for Heidegger that one such type of revealing, which he dubs "technological", reigns supreme. With such an understanding comes the inherent risk of blocking access to any other ways for Being to reveal itself and to forever ward off access to the path of authentic questioning guided by a pious and responsible attitude [3] (p. 26f).

Similar to Husserl's *Crisis*, Heidegger returns to the birth of Western philosophy. But instead of finding Ancient Greek thinking as a beacon of hope which serves as the guiding but perhaps never reachable goal, Heidegger argues that philosophy has always been on an aimless wandering from which it never regained its proper footing. The tradition immediately misunderstood how the basic philosophical question, that of Being (*Sein*), should be posed. Instead of turning to Being as the ultimate precondition, it has instead typically been reduced to a question about beings (*seiend*) and thereby focused on the nature of various types or kinds of entities and how they appear to us[12]. From this follows–according to Heidegger–the privileging of certain modes of existence over others, typically those beings that can be made present to our cognizing faculties. Through the reification of the presence of beings, critical enquiry forgets that its chief task is supposed to be the interrogation of Being as such[13]. As we shall see later on, Heidegger also connects the ultimate forgetfulness of Being to a more general decline of Western thought and culture into a state of anthropocentric relativism and nihilism. A key element in this presumed civilizational decline is a technological mode that renders all values and all (existential) meaning relative to the immediate instrumental concerns of human beings.

Before turning to Heidegger's philosophy of technology and the place of "representations", let me attempt to tease out how this "forgetfulness of Being" (*Seinsvergessenheit*) is approached. The way to even start making sense of Being is claimed to be through the type of being for which Being is a concern: human beings (or *Dasein* as it is also known in Heidegger's admittedly quite convoluted terminology). This analysis culminated in the monumental *Sein und Zeit*: an unfinished preparatory work aimed at laying the ground for adequately asking the question of Being in at least a less misinformed manner. The approach is perhaps best described as a hermeneutic phenomenology that aims to lay bare some of the most fundamental kinds of meaningfulness (or "care", *Sorge*, [26] p. 225f) that characterize Dasein's existence. Notably, the analysis Heidegger proposes remains skeptical of abstracting and overly intellectualizing away from our everyday concrete encounters within the world, and instead emphasizes that we always find ourselves "thrown into" (*geworfen*) a pretheoretically meaningful and unthematically comprehensible context [26]. One famous example of such an understanding concerns the nature of material objects. Instead of conceiving of them in a distanced and reflective mode as having, for instance, a particular material configuration or spatial extension, their authentic mode of disclosure for Dasein is against the backdrop of their practical functions as a "tool" (*Zeug*). A hammer, to use Heidegger's own example, is not primarily an object composed and configured in a certain manner; rather, we encounter it as something that can be adequately used for hammering. To reflect on the essence of hammers or of material objects in general then amounts to a derived form of engaging with our surroundings, but one that nevertheless has come to be philosophically privileged. To retrospectively read the later Heidegger into his early work, we could say that when things are experienced as tools, they resist attempts

to deprive them of their integrity as independently existing things. To then return us to Dasein's basic mode of "being-in-the-world" (*in-der-Welt-sein*) requires a transformation into such a hermeneutic perspective[14].

Heidegger's work from the late 1930s is often described (and also by himself) as a "turn" (*Kehre*) away from both *Being and Time* and his connections to national socialism[15]. While there is a pronounced difference in *how* to do philosophy, his thinking after *die Kehre* nevertheless retains a considerable similarity in *what* the ultimate task is supposed to be. The point where his later thought differs most radically is the shift toward the epochality of Being itself. Instead of seeing Being as something that has eluded philosophy since its inception, it now becomes something subject to variation across distinct historical eras: "[m]etaphysics grounds an age, in that through a specific interpretation of what is and through a specific comprehension of truth it gives to that age the basis upon which it is essentially formed" [3] (p. 115). Certain interpretations where "Being's modes of coming to presence" (*Wesensweise des Seins*) differ lead to distinct phases in the epochality of Being, which remain, in a sense, incommensurable with one another. The task then changes into approaching how the appearance of Being is historically conditioned.

A recurrent reading considers the epochality of Being as a kind of relativism: what something is and what is true vary from a particular time or place. Such an interpretation is strongly endorsed by for instance Foucault's appropriation of Heideggerian motives in *Les Mots et Les Choses* [27] and explicitly stated by Carr [11]. I do, however, read it in a different way. There is only one Being and not different Beings revealed across different historical epochs. What changes is *how* Being comes into our field of understanding in different manners [3] (p. 38). This requires a brief and–admittedly simplified–mention of Heidegger's understanding of truth. Instead of thinking of truth as a relation that can pertain between statements and state of affairs, its most basic and authentic form involves "unconcealedness" (*Unverborgenheit*, see note 2)[16]. Truth is to open up, to bring what was once veiled into presence. But to bring something into our field of understanding is *also*, as Lovitt writes in an explanatory note, to conceal "the source and foundation of all unconcealedness or truth" [3] (p. 36). To fully uncover something and let it be available for examination is to simultaneously forget and hence conceal that *it was previously veiled*. Such basic forms of disclosure are various manifestations of Being where the interplay between concealment and unconcealment differs between various epochs.

In a series of texts, primarily *Age of The World Picture* and *The question concerning Technology* (in [3]), Heidegger argues that modernity is dictated by one such form with a hitherto unsurpassed domination. This is a "technological" mode of understanding where the world is directly connected to the appearance of certain manners of representing it. Just as for Husserl, it is important not to read this historical account as an investigation into factual history but as an enquiry into the epistemological preconditions that enabled certain ideas or technologies to flourish at a particular time.

*4.2. Gestell and Bestand: Modernity as the Age of "Machine Technology"*

A quick look in the *Cambridge Dictionary* finds *technology* defined as "(the study and knowledge of) the practical, especially industrial, use of scientific discoveries"[17]. The artifacts these practices create are typically thought of as having no agency independent from their use. When we talk about their possible dangers, it is in terms of the effects from a careless and mindless application ignorant of the risks. They might have apocalyptic ramifications (like nuclear war) or risk hollowing out some perceived essential feature of humanity, but this is to a large extent dependent on how we chose to use them. It would then be our collective responsibility not to succumb to the dangers that can be set forth by technology. Heidegger [3] explicitly dismisses such criticisms of technology as naïve and ultimately harmful. The "essence" of technology, he claims, is not something technological per se. It cannot be found as a property common to artefacts like coffee makers, particle accelerators, smartphones and AI-controlled kamikaze drones. Neither is the technological found in the scientific knowledge that made such artefacts possible. Instead, its essence

can be found in a specific form of disclosure characteristic of modernity [3] (pp. 12f, 23f). From such a point of view, to assume the neutrality of technology amounts to "danger as such" [3] (p. 26).

What would be so specifically threatening with technology and so perilous with considering inanimate things as attaining their meaning through their application? Heidegger begins, as mentioned, by claiming that (the essence of) technology is a form of revealing: a way the world appears in the modern age; or to phrase it differently, technology is an epochally defined disclosure of Being. In modernity, this form of revealing is unquestioned and remains for all practically relevant intents and purposes unchallenged. With its immense and careless domination of the world comes the ultimate risk that the possibility for other ways to disclose the world will irredeemably be lost [3] (pp. 27, 30f, 33f). As this occurs, Heidegger finds (in a vein clearly reminiscent of Husserl's critique summarized in Section 2) that a modern conception is at risk of extinguishing the search for meaning and an authentic encounter with the world. There might still be instrumental progress, but this it deemed as an improvement against a background already presupposing the validity of a technological mode of revealing.

We have so far touched upon why technological revealing is harmful, but we still need to clarify *what* it is. One way to understand this is through its revealing of nature. In contrast to other forms of revealing, technology makes nature available for human beings to use in any way we see fit. Its unconcealment is in the form of a "standing-reserve" (*Bestand*): a resource for domination and exploitation for instrumental purposes according to detached calculations. Nature is thus transformed into something with an instrumental value as an everywhere conveniently available repository.

> The earth now reveals itself as a coal mining district, the soil as a mineral deposit. The field that the peasant formerly cultivated and set in order [*bestellte*] appears differently than it did when to set in order still meant to take care of and to maintain. The work of the peasant does not challenge the soil of the field. In the sowing of the grain it places the seed in the keeping of the forces of growth and watches over its increase. But meanwhile even the cultivation of the field has come under the grip of another kind of setting-in-order, which sets upon [*stellt*] nature. It sets upon it in the sense of challenging it. Agriculture is now the mechanized food industry. Air is now set upon to yield nitrogen, the earth to yield ore, ore to yield uranium, for example; uranium is set upon to yield atomic energy, which can be released either for destruction or for peaceful use. [3] (pp. 14,15)

For nature to be unveiled as a standing reserve, a corresponding form of revealing is required. Heidegger labels this with an idiosyncratic use of the German word *das Gestell* ("enframing")[18]. To be enframed means to be deprived of integrity and independent existence and to be fully determined by the manners in which it can be deployed as a reserve. As enframed, Rhine is no longer primarily a river: it is a standing-reserve for producing electrical power.

> The hydroelectric plant is not built into the Rhine River as was the old wooden bridge that joined bank with bank for hundreds of years. Rather the river is dammed up into the power plant. What the river is now, namely, a water power supplier, derives from out of the essence of the power station. [3] (p. 16)

The matrix *Gestell-Bestand* can thereby be seen as the precondition for the existence of technological artefacts in the first place. It is the "ground plan" (*Grundriß*) essential for seeing nature as fully exhaustible by the ways in which it can be exploited for our purposes. Within this mode of revealing, things are deprived of their independence and integrity, turning their primary value into a resource for our purposes [8] (p. 165f). Things are, Heidegger writes, treated with an "injurious neglect" (*Verwahrlosung*, [3] p. 48).

In the technological mode, values are relationally determined through enframing. It is here important to remember that Heidegger is not putting the blame on individual

persons. He is not trying to burden the reader with eco-anxiety or flight shame. Whether we want it or not, human beings of today are put [*stellt*] in the position to enframe nature as a standing reserve. We are not free to choose to disclose the world in a different manner. In the technological age, man for his part is already challenged to exploit the energies of nature [3] (p. 18). As more and more of nature is turned into a reserve, with a corresponding affirmation of an enframing kind of revealing, our understanding of the world also loses the possibility to free itself from this particular form of understanding. In the end, "[t]his illusion gives rise in turn to one final delusion: It seems as though man everywhere and always encounters only himself", Heidegger writes [3] (p. 27). The potential of enframing is without limits: *anything* can be subjected to such a productive machinery. However, it is important to recall that human beings are dominated by enframing rather than the other way around. This means that there is nothing that stops it from turning human beings into a standing-reserve, which would mean a radical abdication from human beings as Dasein with the responsibility to serve as the "shepherd of being" [26] (see Section 5 for a discussion on whether the epochal manifestation of Being is impenetrable) While we are not (yet) at the point where people are turned into Soylent Green, we still find the seeds of such conceptualization embedded in notions like "human resources" [3] (p. 18). Heidegger thus sees this transformative capacity as one of the inherent risks with technology: nothing escapes the exploiting logic of enframing that sees everything as an available standing-reserve[19].

### 5. Heidegger on "Representations"

One of the more radical conclusions to be drawn from Heidegger's analysis is the heterogeneity of Being across different epochs. It comes to our understanding in incommensurable ways dependent on the historical context we find ourselves "thrown" into. While human beings do in a sense participate actively in maintaining a particular form of revealing, Heidegger also persistently states with a resigned acceptance that the epochal disclosure of Being is nothing that can be overcome. It is out of our hands to affect and it permeates all aspects of human lives. How Being comes to be revealed cannot be changed by an act of willful determination: it is a "decision" that has always already been made for us and not by us.

One might be inclined to object that Heidegger himself seems to be able to make sense of revealing in epochs other than his own. Not only is he able to cross the barriers between allegedly impenetrable historical eras, but he passes through the mists of technological revealing claimed to be completely dominant in modernity. If the epochality of Being is indeed impermeable, then the Heideggerian reading of history should be impossible. But since he is overcoming this hurdle, it seems to prove that Being is not completely determined by a specific set of historical conditions. Heidegger does not discuss whether a panhistorical outlook is presupposed, but I take him to mean–in contrast to the reading seemingly endorsed by Foucault–that there is indeed a possibility for communication across epochs that can also allow for tracing the genealogy of each epochal disclosure. In the case of technological revealing, Heidegger insists that it does not just come into existence *ex nihilo*, but its genesis can clearly be sought in the origin of natural science and specifically by the role representations (*Vorstellungen*, a word also derived from the German verb *stellen*) play in its establishment. In this manner, the disclosure of Being as *Bestand* forms in history through an unquestioned and passive acceptance of "objectivity". What makes modernity such an existential threat is its domination characterized by a pseudo-activity (*Betrieb*) reminiscent of Husserl's description of more and more facts based on ever refined calculations and measurements without any reflection on the meaning of this activity [1] (see Section 2).

Heidegger's view on the privileged and peculiar role of representations in a technological age specifically involves a creative reading connected to his overall analysis of modern culture[20]. Taking a step back, we can ask how something, anything at all, can come to be seen primarily as a standing reserve. When such a mode is dominant, the

world is one where things are placed as conveniently available for domination and use, which also means that they are deprived of their integrity as independently existing entities. What they are is exhausted by their instrumental relation to human purposes. What forms the horizon of such a utilitarian-calculative rationality where entities are determined by relations of predictable manipulation? Heidegger argues that the primacy of instrumental value involves the ambition to gain control and mastery over its activity. It cannot be left to its own accord or to chance, but it must be tamed by forming expectations in advance and providing explanations in retrospect [3] (p. 121ff). These predictions must also be of such a character that they allow for control to be repeatable with exactitude: a way to devise mathematical formulae and apply them in precise measurements, which means a rigid method for calibrating with exactitude. At such a stage, we are no longer operating directly with the thing in question, but comprehension is transformed into one mediated by models and abstractions assumed to capture an entity in its essence. This approach, which could be seen as a basis of any scientific procedure, thus involves a distancing whereby a thing is understood in terms of how it is represented in a manner that allows for manipulation and control. While Heidegger does not phrase it in such terms, there is a parallel to Husserl's insistence on natural science as cloaking the life world from which it once originated. On this specific point–and limited *only* to their respective critique of modernity–the recurrent Heideggerian remark of truth as an interplay between unveiling and veiling reminds us of Husserl's point that Galileo's naturalistic breakthrough is simultaneously an act of revelation and of dementia.

Heidegger proposes that the type of understanding inherent in exact mathematical models are essential prerequisites for objectivity to emerge in the first place [3]. There are two quite different types of representations that Heidegger connects to modernity: mathematically precise measurements and the aspiration to represent the world in its totality, which Heidegger calls "world picture" (*Weltbild*). Exact representations can be expressed in different ways, but they do so with the intent to be rigorously precise and mathematically definable. As Heidegger sees it, the claim for exactitude will participate in determining what is real, which Pattison summarizes in the following way: "[m]athematics is projective, in that it runs on ahead of actual experience, determining in advance and entirely in terms of its own self-determining laws what can and cannot count as knowable" [28] (p. 4). The belief in the possibility for transparently capturing the primary features of objects in precise and accurate measurements is central to natural science, and hence to a certain understanding of objectivity. With the help of these representations, it becomes possible for human beings to manipulate and control their environments in a predictable fashion. As Heidegger sees it, this "injurious neglect" of the integrity of things is an inextricable component in the emergence of technological revealing. An undisputed faith in "scientific projection [as] determining its object" [28] (p. 8) prepares the way for enframing. It implies, as Heidegger provocatively suggests, a direct correlation between objectivity and subjectivity. The former is attained by the exactitude of scientific explanation, which means a corresponding rise in the subject as ultimately serving as the arbiter of representational validity [3] (pp. 128, 132). Such an interpretation of subjectivity, as not only historically conditioned but also intertwined with objectivity, is clearly a point where Heidegger's reading is completely irreconcilable with the basic tenets of Husserlian phenomenology.

At this juncture, it should be noted that Heidegger is far from clear exactly "representation" means, which also seems to have eluded previous commentators (cf. [29] p. 184). In a sense, Heidegger's silence on the exact contribution from our attitude towards semiotic structures and representational technologies in the formation of objectivity is reminiscent of Husserl's lack of clarity on how language preserves ideality in sedimented structures (see Section 3). To fully do justice to the complex nature of language and other semiotic systems in Heidegger's thinking, including the often quoted but ultimately mystic remark that "language is the house of Being" [30] goes far beyond the scope of this paper (and admittedly, my intellectual abilities). Let me just mention a few key points on the use of the term "representation" (i.e., as a translation of Heidegger's term *Vorstellung*) of specific

relevance in the present context. It is abundantly clear that 'representation' for Heidegger is not to be read as synonymous to signs as a unity comprised of expression and content (as noted in note 17). Instead, his use seems crucially to emphasize how representation involves forming an idea of something and attempting to capture it in a manner that does not leave it to its own accord. A specifically modern conception is that things are most adequately comprehended by descriptions relying on exactitude. As I read this, Heidegger sees these representations as insidiously disturbing our ordinary encounter the world. The experience of a thing is intervened by the presupposition that its ultimate existence is exhaustively described and precisely measured in a manner possible to preserve in external form and implement in various forms of technological artefacts. In this way, our idea of objectivity involves formal mediating structures assumed to be more truthful to the ultimate nature of reality than the things they supposedly stand for. They are so by virtue of operating with an already pre-defined blueprint for reality as being measurable and quantifiable. It is on the basis of such abstractions, idealizations and models that things are placed in a way completely alien to what Heidegger takes to be the "ordinary", even if it remains quite unclear what amounts to such an ordinary way of experiencing. Similar to Husserl's analysis of formalization and technization as participating in the formation of the crisis, these structures are for Heidegger nevertheless assumed to be–due to their exactitude and repeatability–more real than the things themselves.

An example might bring some much-needed clarity and concretion. Speaking of the Newtonian understanding of a physical body, Heidegger claims that "there is no such body. There is no experiment which could bring such a body to intuitive representation [*anschauliche Vorstellung*]" [31]. As we learn in physics and chemistry, an object is not actually the thing we take it to be, but it is comprised of atoms in various molecular configurations (which in turn is further composable into subatomic parts). Such scientific knowledge of these presumably real things is both conceptually and practically made possible by and preserved in what Crichton [29] calls "modern representations" and "representations against experience". These are what allow for further scientific explorations of the natural world, which by their very nature can give birth to even more precise calculation in the "ongoing activity" of science. Importantly, these representations do not just stand-in for reality but are for Heidegger deviant from our normal understanding of the world[21]. As they do so, they also interfere with our authentic encounter whereby "objects" are created out of "things" (e.g., [8] p. 163f; see also the discussion of tools in Section 4). It is by this transformation of the ordinary into the calculable that Being is conceived as explicable by means of what can be exactly captured. Within such a mindset, what would be intuitively comprehended as "the real things" become hidden behind a veil of exactitude. Heidegger thus argues that while science can offer a true form of revealing, it is ultimately reductive by depriving things of all types of properties that cannot be exactly measured. These representations involve precise and calculated idealizations that force things through the narrow corridor of technological enframing. Here, they are all given the same treatment assumed to be exhaustible for capturing what they are. Objects are defined by how they are measured and quantified, which is the only type of understanding that such a mode of understanding can allow for. The place for representations is thus to let us capture things as "objects" for us rather than things with an independent existence, culminating in totalizing and all-encompassing representations of the "world" itself, which Heidegger [3] (p. 115f) dubs "world picture" (*Weltbild*). The belief that the world in its totality can be truthfully and adequately represented serves as the ultimate expression of the hubris found in the objectifying gaze that Heidegger links to modernity.

Despite his extremely negative attitude towards science, Heidegger nevertheless insists on its validity [3] (pp. 167–168). Technology *is* a way for Being to manifest itself. His critique seems to question its dominance and a perceived lack of a counterweight to modernity. All the more surprisingly, shockingly even, is that Heidegger also expresses a sudden optimism towards technology. At the end of *The Question concerning technology* [3] (p. 34), he cites the following two stanzas from Hölderlin's poem *Patmos*:

> where danger is, grows
> the saving power also
> (wo aber aber Gefahr ist, wächst
> Das Rettende auch)

While this surely seems like an unmotivated change of heart, I assume that Heidegger wants to point to the importance of art for letting us in some manner draw closer to Being. Art (including poetry) and technology are both covered by the Greek word *tékhnē (τέχνη)*, a unity concealed in modernity. In the form of the artistic expression, Heidegger finds the possibility to avoid falling into traps of an objectifying representational gaze (see [3] for technology and art and [32] for an extensive discussion on Heidegger's conception of art and its connection to truth). On the other hand, to privilege a mode of language that does not even try to fixate sense in a systematic way would be completely anathema to Husserl. Instead of turning away from such ideals as truth and knowledge as they are usually conceived, Husserl consistently strived for their improvement and phenomenological rehabilitation–a case in point being his insistence on the responsibility of the community for combatting the worst effects of sedimentation (see Section 3 above and [2] pp. 362, 372).

## 6. Conclusions

The main goal of this paper has been a comparative reconstruction of both the similarities and differences of Husserl and Heidegger's respective critique of modernity. In doing so, I specifically argued that they–albeit in radically different ways–connect their critique to semiotic matters, and specifically to the type of formalized structures assumed to be necessary for objectivity. Despite the enormous power that such formalizations have been shown to have, Husserl and Heidegger both level a critique against the way such an approach has interfered and intervened with our understanding of the world and ourselves. Instead, the insistence on the unquestionable validity of signs and representations conceals their own origin and manner of disclosure. In this last section, I wish to briefly discuss two issues that have been somewhat implicit so far. First, is science that dominant in contemporary culture and second are the semiotic structures identified by Husserl and Heidegger unique to modernity?

One might argue that the risks of scientific and technological domination has turned out to be an ill-founded fear. Carr [11] suggests that the true victor in 20th-century philosophy was neither phenomenology nor naturalism, but relativism (which Carr exemplifies with thinkers like Wittgenstein, Foucault, Derrida–and Heidegger). While this observation is certainly correct, it seems that such constructivist and relativist perspectives are mainly dominant within particular domains of contemporary society, such as humanistic departments of Western universities. In a broader perspective, I think we can find ample indications of a reliance on a technized scientific mindset in our present times–these have become so entrenched and well-established that they hardly seem questionable anymore. Let me just mention two examples: technocratically based decisions-making that actively engage scientific expert knowledge for issues concerning political and economic issues, and the application of formal quantifiable procedures to human life and culture. Such processes of "commensuration" do however qualitatively transform the phenomena they are supposed to quantify [20]. To what extent these changes are considered to be an improvement or not is, I gather, a matter of ideological conviction. I do however believe that neither Husserl nor Heidegger would claim their neutrality. Moreover, Carr does not connect his discussion to Husserl's implicit connection between skepticism and scientific positivism (see in particular [1] (p. 86f). Once the contribution of the life world and transcendental subjectivity has been cloaked by a mathematization of nature, the inclination is to approach human beings in a similarly naturalized manner. As this occurs, Husserl seems to suggest that the unstable footing of modern science invites a Humean type of skepticism that finds both scientific concepts like causality and subjectivity to be ultimately fictious constructs.

Concerning the connection between modernity as harboring both naturalism and skepticism, Heidegger argues in a similar vein that the "history of metaphysics" as cul-

minating in anthropocentrism is clearly articulated by technological enframing where everything is evaluated in relation to its instrumentality as a standing reserve. Such an anthropocentric-relative basis will inevitably make all values hollow, culminating in a nihilism of a Nietzschean "will to power" (e.g., [33]).

The second issue concerns exactly how modern these "modern representations" actually are. As I noted in Section 3, it would seem as if Husserl's analysis of sedimentation presupposes that the effects of sedimentation are at least principally operational *even on the first mention*. One might then ask (as for instance Derrida [24], Lawlor [34,35] and Buckley [4] do) whether Husserl is pointing to something specific in modernity, or if he is beginning to detect a general feature that regulate linguistic communication as always drifting away from the authentic sense it seeks to express and preserve. In a similar manner, Heidegger's insistence on modern representations as involving precision that conceptually contribute to turning things into a standing reserve can be seen as the outcome of a long historical process. Ancient civilizations measured and manipulated things, with at least an aspiration towards exactitude and repeatability. The careless destruction of the environment or strife for efficacy in warfare did not follow modern technology, but rather seems to be the other way around. A possible synthetic reconciliation that–as is often the case with syntheses–might make no one happy is to say that these semiotic structures are not unique to modernity, but that they reach a peak height by gaining an independent momentum and breaking free from the world that once gave birth to them. Such a reading would nevertheless need to be complemented by a much more careful analysis of the semiotic properties involved in the different types of signs and representational technologies than what could be discussed here (mathematical symbols, diagrams, writing abstract models, world pictures, etc.). In the case of Husserl, a separation between different types of written signs and a nuanced discussion of how formal signs are properly grounded is required (see, for instance [19,22]). In the case of Heidegger, the difference between intuitive and modern representation remains unclear without their respective semiotic differences specified. To fulfil the ethos of contemporary academic publishing, the promise of such a more comprehensive analysis must however remain postponed and filed under "future research".

**Funding:** This research received no external funding.

**Acknowledgments:** A huge thanks to two anonymous reviewers for providing important references and suggestions for clarifications. I am very grateful for the meticulous reading and—admittedly—quite challenging comments from the guest editors Jordan Zlatev and Göran Sonesson. By engaging with their critical reading, the paper has improved considerably. Finally, John Haglund's feedback on an earlier draft was invaluable in giving the paper a clearer direction.

**Conflicts of Interest:** The author declares no conflict of interest.

## Notes

1   Put briefly, Derrida would question whether it is ultimately possible to locate sense "outside" of semiotic structures. Instead, such a prospect may be endlessly deferred by further chains of signification.

2   While the English translations use "concealment", the original German words have quite different connotations with respect to the manner and intent behind which something has become concealed: Husserl uses *Verdeckung* by [1] (pp. 52, 53ff), which suggests that something has been hidden and covered over. Heidegger's term *Verborgenheit* [3,8] has connotations to something concealed in a secretive and mystical manner. Importantly, *Verborgenheit*—together with its negation *Unverborgenheit*—are crucial not just to a criticism of modernity, but also for Heidegger's alternative conception of truth as "unconcealment" (see Section 4).

3   A concrete example of how Husserl may have thought of science as having lost its way is the abuse of science in such warfare technologies like machine guns and poisonous gases during World War 1, see for instance [4,12].

4   For in-depth accounts of Husserl's conception of the crisis, see [4].

5   Husserl seems to be inspired by Herodotus' and Aristotle's claim of geometry as being borne out of the redistribution of arable land after the seasonal flooding of the Nile. Schemmel's [36] detailed historical overview of the development of space is roughly similar to the general outline of Husserl's account, but differs concerning the historical details.

6　Husserl's characterization of geometry in *Crisis* are somewhat implicitly relying on his previous analyses of spatial constitution, e.g., [16]. Taken together, the picture that emerges is the recognition of an increasing abstraction in several intermediary steps that ultimately culminates in the type of pure ideality that Husserl finds exemplified in geometry. What characterizes such an ideality is its independence from the concrete life world: a notion like 'circle' is not just a more abstract concept of 'round' but one that can only be imperfectly and approximately materially instantiated or represented.

7　The place of history in Husserl's account is not just an empirical matter where a sequence of events leads to other events. Instead, history, and especially what Husserl calls the "historical *a priori*" [2] (pp. 372–374) takes on an integral and formative function in the transcendentality of the life world.

8　In line with one of the fundamental tenets of Husserlian phenomenology, the scientific process for finding truth is—just as the constitution of any intentional object—principally inexhaustible.

9　See [19] for a nuanced and critical discussion on whether Husserl manages to secure formal categories and uncover their ground in perceptual intentionality.

10　It should be noted that these late reflections comprise just a minor part of Husserl's writing on language and semiotics. It is thereby a matter of controversy and discussion in the literature exactly what to make of them. Some interpretations proposed by for instance Merleau-Ponty, Derrida and Lawlor [24,34,35,37] take these late thoughts to serve as a substantial addition to and possibly even a self-critique of Husserl's previous view on language whereas the opposite viewpoint is defended by for instance [12].

11　Husserl's understanding of ideality is far more complicated than what I can do justice to in this context. As [24,35,37] among others have noted, the survivability of ideality seems to necessitate its material instantiation in the life world.

12　To indicate the so-called "ontological difference" between Being as such and beings, Heidegger relies on a distinction in German between the verb to be, *Sein*, and its present participle *seiend*. This is usually indicated in English translation with *Being* vs. *being(s)*.

13　The forgetfulness of Being has some parallels to Husserl's analysis of the crisis, but is clearly more radical in locating the oblivion already at the birth of Western philosophy. See [4] for insightful comparisons.

14　Heidegger's explicit skepticism and ultimate rejection of an overly intellectualized attitude is often read as a critique of Husserlian phenomenology. Successfully or not, it could however also be seen as an attempt to continue delivering on the ambition of returning thinking to its source in the life world.

15　With respect specifically to "*die Kehre*" as an attempt of Heidegger to distance himself from national socialism, see [38].

16　In his interpretation, Heidegger relies on the etymology of the Ancient Greek word *aletheia* (ἀλήθεια) for truth, which can be analyzed as literally "not-hidden".

17　https://dictionary.cambridge.org/dictionary/english/technology (accessed on 14 October 2022).

18　The terms *Gestell* and *Bestand* do not translate well into English. The standard translations "enframing" and "standing-reserve" do not preserve their derivation from the highly productive German verb *stellen* with meanings such as 'to set in place' or 'to set upon'. Due to its fundamental importance in Heidegger's analysis of the modern age, its morphological productivity and the associated lexical fields should be kept in mind in the following. One should especially be aware of *stellen* as expressing activities like ordering (both in the sense of requesting something and giving a directive), arranging, making available for instrumental use, and regulating.

19　In a sense, one could even argue that the space for contemporary political decision-making is fundamentally caught up in the logic of *Gestell-Bestand*. This is perhaps nowhere as apparent as in discussions of climate change where the proposed solutions seem more often than not to be of the same kind that led to the present state.

20　A note on the difficulties with key terms like 'signs' and 'representation', especially when these occur in translation: I use 'sign' as the general term for whatever conveys meaning as a structure of expression and content (as for instance [39]). Following standard translation, I use 'representation' for Heidegger's *Vorstellung*, which point not just to signs but also to a particular objeticfying attitude towards their validity to capture things more truthfully than the things themselves (see below).

21　In discussing objectivity as being borne out of conceptions of importance for enframing, Heidegger draws attention to the fact that the German word for 'object'—*Gegenstand*—has the same root as *Bestand*, cf. note 18 above.

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
