# Peer review of "Husserl and Heidegger on Modernity and the Perils of Sign Use"

_philosophies, doi:10.3390/philosophies7060120_

Round 1
Reviewer 1 Report
This study (“Husserl and Heidegger on Modernity and the Perils of Sign Use”), in my opinion, is clearly a good one, and it is absolutely worth publishing. It is a comparative study of Husserl’s and Heidegger’s account of modernity and its crisis, with a special regard to the problem of signs and language. Both Husserl and Heidegger say that there are fundamental problems with modernity, and both late Husserl and Heidegger claim that these problems have a particular connection with language in general and the language of modernity and modern science in particular. The crisis of modernity is also a crisis of its language. In Husserl, the formal language of modern science is separated from its original sources, from the life-world, and ultimately from constituting transcendental subjectivity. In Heidegger modernity and modern science in particular – alongside with its language – separates us from Being, it alienates us from the latter, and reifies it.
The author treats these features in her/ his study in great detail, and – in my view – in a very convincing manner. This essay is also written in a good and smooth English, I believe it is really a pleasure to read, and it is also a very informative text.
I think that minor improvements are still possible, but I don’t want to enforce anything. I believe, that this study could be published in its present form. It’s informative, it has a clear train of thought, and it is based upon a far reaching knowledge of primary and secondary literature.
I also liked very much the pop-cultural references in the text – e.g. “Soylent Green” on page 12!
1) The presuppositions of the two philosophers could have been made more explicit. Husserl is obviously committed to the project of refounding (Neubegründung) of modernity and modern science. He is clearly committed to the endeavour of Enlightenment and its enthusiasm about rationality. Husserl thinks that something obviously went wrong with modernity, but he also believes that this project is not inherently or essentially a failure. He just says that one must keep open or disclosed the original sources of meaning, and the project could be placed on the right track with it.
Heidegger, on the other hand, less explicitly (there are places where he even denies that) but have some clearly demonstrable affinities concerning German Romanticism. He has really, really strong reservations against modern rationality and also theoretical attitude in general. He thinks that they just alienates us from Being, modern rationality and science reifies the Being, and the whole project of modernity is nothing but a failure that leads all of us towards a global, even Being-historical (seinsgeschichtliche) catastrophe. Other than Husserl, he thinks that modernity, even rationality and theoretical attitude, cannot be corrected.
Heidegger – as he clearly puts it in e.g. his Letter on Humanism, but also other late writings – believes that poetry and poetic way of thought (dichterisches Denken) is the authentic relationship to Being, while the modern scientific attitude is the totally inauthentic relationship.
But still I think, in these assumptions, Husserl’s positive affinity towards Enlightenment, and Heidegger’s positive (but not so explicit, not so honestly declared) affinities towards German Romanticism (especially towards Hölderlin) are still crucial in this context.
2) I was little surprised that the author, despite the hints regarding the environmental, ecological aspects of the problem, there are no references in the text to the tradition of eco-phenomenology, which is a quite important and relevant movement in the context treated by the author.
Regarding Husserl, the author might find we useful perhaps the study of Erazim Kohák: “An Understanding Heart” – which, at certain points, has very similar considerations and inferences like what the author does.
Regarding Heidegger, I would like to draw her/ his attention to the Heideggerian ecological tradition in America, especially to the works of Michael Zimmermann, Leslie Paul Thiele and Frank Schalow. For the topic, treated in the articles, these authors could be utmost relevant.
Reviewer 2 Report
“Husserl and Heidegger on Modernity and the Perils of Sign Use” presents an analysis of the “semiotic” functioning of writing, mathematical formulae and diagrams in Husserl’s and Heidegger’s accounts of the conceptual and correlative ontological shift operative in the early modern science of nature. Husserl’s account of modern mathematical physics’ self-interpretation of its institution of a symbolic method as the in fact true ontology of nature and Heidegger’s account of the Gestell (enframing) of the meaning of Being brought about by the mathematical projection constitutive of modern mathematics are compared. The comparison is made with a view toward the deficits in the original meaning of the world (Husserl) and Being (Heidegger) that the author argues characterize Husserl and Heidegger’s accounts of the limits of modern science. Specifically, the author argues that each thinker maintains that the exactness of objective meaning made possible by modern mathematics combined with the utilization of this mathematics in the service of the measurement of nature necessarily functions to conceal the more (and in some cases the most) original sense and meaning of the being of the world (Husserl) and the meaning of that being (Heidegger).
Despite the condensed presentation of its comparison and analysis, the paper in general is quite clear and as far as its argument goes, compelling. However, on my view the overall argument is limited by its lack of recognition of the difference Husserl makes between generalizing and formalizing universality and Heidegger’s appropriation of this difference, in its semiotic analysis of the limits of sense and meaning in those two thinkers’ account of modernity. The question posed at the end of the paper, whether the limits on the disclosure of original meaning is a function historically dated to the employment of written signs in modern thought or whether it’s a limit of the graphic embodiment of meaning per se is unnecessarily and therefore misleading limited because the semiotic differences in the expression of generalized and formalized universality are not addressed in the author’s analysis. This distinction is especially important for the author’s analysis, given both Husserl’s and Heidegger’s agreement that formalized universality is a particularly modern innovation, and therefore one with no structural and historical precedent in pre-modern concept formation. A discussion of these differences can be found in: Husserl, Ideen zu einer reinen Phänomenologie und phänomenologischen Philosophie, I, Buch: Allgemeine Einführung in die reine Phänomenologie, hrsg. Karl Schuhmann (Hua III), Den Haag 1976, 26–27; Heidegger, Einleitung in die Phänomenologie der Religion, in Phänomenologie des religiösen Lebens, GA 60 (Frankfurt am Main: Vittorio Klostermann, 1996), ch. 4; Jacob Klein, “Phenomenology and the History of Science,” in Philosophical Essays in Memory of Edmund Husserl, ed. Marvin Farber (Cambridge, Mass.: Harvard University Press, 1940), 143–163; reprinted in Jacob Klein, Lectures and Essays, ed. Robert B. Williamson and Elliott Zuckerman (Annapolis, Md.: St. John’s Press, 1985), 65–84; and Burt Hopkins, “Deformalization and Phenomenon in Husserl and Heidegger,” in Heidegger, Translation, and the Task of Thinking: Essays in Honor of Parvis Emad, ed. Frank Schalow (Berlin: Springer, 2011), 49-69.
Given the importance of the phenomenological difference between generalizing and formalizing universality in general and its relevance in particular to the author’s topic, I recommend that a necessary condition for its publication be that the author expands the paper’s analysis to take into account its significance for both that analysis and its conclusions.
Finally, there’s one sentence in the abstract that contains a phrase I do not understand. The sentence: “However, the demand for objectivity, exactitude, and repeatability insidiously interferes with the meaning that such representations seek to express, which is a duality of objectivity encapsulated in ‘the sedimentation of meaning’.” The phrase: “which is a duality of objectivity encapsulated in ‘the sedimentation of meaning’”. I suggest that the author unpack what this phrase intends to express for the benefit of the reader.
Round 2
Reviewer 2 Report
I have reviewed the revised version of the manuscript and in my assessment it has been sufficiently improved.
Author Response
Dear Editors,
I have revised the paper according to the suggestions. New changes are marked in GREEN.
All the best,
Johah
